# Quality and Yield of Bell Pepper Cultivated with Two and Three Stems in a Modern Agriculture System

**Jorge Flores-Velazquez** [1,*] , **Cándido Mendoza-Perez** [1,*] , **Juan Enrique Rubiños-Panta** [1] and **Jesus del Rosario Ruelas-Islas** [2]

1. Hydrosciences, Postgraduate Collage, Campus Montecillo, Carr Mex-Tex Km 36.5, Texcoco Edo de Mexico 56230, Mexico
2. Faculty of Agronomy of the Valle del Fuerte, Univerdity Autonomouds of Sinaloa, Sinaloa 81110, Mexico
* Correspondence: jorgelv@colpos.mx (J.F.-V.); mendoza.candido@colpos.mx (C.M.-P.); Tel.: +52-777-2588-956 (J.F.-V.); +52-551-3528-662 (C.M.-P.)

**Abstract:** Bell pepper is a very important crop for its value in domestic and foreign markets. Actually, growers have adopted different management practices. In that aspect, management with different numbers of stems can define the quality and quantity of the product, as well as any increase in yield. The objective of this work was to evaluate the physical and chemical characteristics of fruits in terms of the quality and postharvest of bell pepper, as well as the yield according to the number of stems grown in a hydroponic system under greenhouse conditions. The experiment consisted of four treatments: two stems (T1) and three stems (T2) on a 'Cannon' cultivar, as well as two stems (T3) and three stems (T4) on a 'Bragi' cultivar. Fruits were sampled to determine total soluble solids (TSS), titratable acidity (TA), pH, electrical conductivity (EC), maturity index (MI), vitamin C (VC), lycopene content, size, shape, color, firmness, and yield. High vitamin C concentration of 120 mg 100 g$^{-1}$ was found in both cultivars. Treatments from 'Cannon' had the higher TSS content, lycopene levels and firmness. Regarding the physical characteristics, T1 of 'Cannon' had better fruit size: 63% (large), 35% (medium) and 2% (small). The highest yield was obtained in T2 of 'Bragi' with 6.50 kg m$^{-2}$. It was observed that total number of fruits increased as the number of stems increased. However, the size of the fruits decreased.

**Keywords:** *Capsicum annuum* L.; physical characters; quality; biochemical components; hydroponics

## 1. Introduction

Mexico is one of the major producers of bell peppers under greenhouse and open field conditions, with most of the production being for export. Approximately 5800 hectares are planted throughout the country, and the yield gap ranges between 80 and 150 t ha$^{-1}$ under greenhouse production and 8 to 43 t ha$^{-1}$ in open field conditions [1]. Recently, the export process to USA and Canada has been increasing, with approximately 240,000 Mg in 2006 [1]. Bell pepper (*Capsicum annuum* L.) belongs to the botanical family of Solanaceae [2], it is native from South America, specifically from region of Bolivia and Peru. It is consumed in fresh, either green or in a more advanced state of maturity.

Bell pepper grown under greenhouse conditions can reach a market value of five times higher than the products of field production due to quality, especially when marketed until the fruits change color (red, orange, yellow, cream, chocolate and purple) [3]. Production of bell pepper in Mexico is organized around a Dutch system type, which consists of maintaining the plant with two V-shape stems by pruning one of the branches so that one fruit grows at each node of both stems. By limiting the number of fruits that grow simultaneously, the system can balance the source–sink relationship so that it is possible to achieve continuous production by keeping the growth throughout the year with plants of more than 3 m long, with a density of two and three plants m$^{-2}$ (four to six stems

$m^{-2}$) [4]. This system demands high production costs, good crop management practices and controlled environments to achieve more than 150 t $ha^{-1}$ [5].

Bell pepper fruits undergo a marked change in color during ripening due to variation in the concentration of pigments (chlorophylls on green fruits), while yellow and red fruits have higher concentrations of lycopene.

Lycopene is a carotenoid mainly found in tomato (*Solanum lycopersicum* L.), bell pepper (*Capsicum annuum* L.), watermelon (*Citrullus lanatus* var. Lanatus) and other fruits or vegetables [6,7]. It also has antioxidant, anti-inflammatory and chemotherapeutic effects over heart–brain diseases and cancer [8]. The color is another physical characteristic that defines the quality of fruit. Temperature, storage time and transport have significant impact on the attributes of physical quality like weight, color and fruit firmness [9].

The number of stems could be an efficient tool to manage quality, quantity and yield of bell pepper (*Capsicum annuum* L.) due to the limited information available on management under greenhouse conditions. As the number of stems per plant increases, the number of fruits per plant is expected to increase. However, the physical and chemical features, and yield can be affected. Therefore, the objective of this work was to evaluate the physical and chemical characteristics of fruits in terms of quality and postharvest as well as the yield according to the number of stems grown in a hydroponic system under greenhouse conditions.

## 2. Materials and Methods

### 2.1. Experimental Site Description

The experiment was established in a zenithal greenhouse, located at the Graduate College on Montecillo Campus, State of México, (19°28′05″ N and 98°54′31″ W of 2244 m altitude). Four treatments of two commercial indeterminate growth cultivars were evaluated: 'Cannon' (red bell pepper) and 'Bragi' (yellow bell pepper), both of which were purchased from Harris Moran seeds® (Davis, CA, USA).

The seeds were planted in germination trays on 14 March, and transplanted on 15 May 2017 with a density of 3 plants $m^{-2}$.

The planting method consisted of the triangular system—'tresbolillo'. Seedlings were placed in black polyethylene bags (0.35 × 0.35 m) with red volcanic sand (tezontle) at 0.4 m apart from each plant, 0.4 m between lines.

### 2.2. Irrigation System

The irrigation system consisted of a watering line (16 mm diameter) with self-compensating drippers (0.4 m apart), a flow rate of 4 L $h^{-1}$ and an operating pressure of 68.64 kPa. Steiner's universal nutrient solution (1984) was applied with an osmotic potential of −0.087 MPa and a pH of 6.5 during the crop cycle. A flow rate of 0.18 L $plant^{-1}$ $d^{-1}$ was applied during the first 30 days after transplant (DAT), 0.480 L $plant^{-1}$ $d^{-1}$ in the vegetative stage, 0.60 L $plant^{-1}$ $d^{-1}$ in the development stage, 0.80 L $plant^{-1}$ $d^{-1}$ in the production stage (peak demand) and 0.72 L $plant^{-1}$ $d^{-1}$ at the end of the season.

### 2.3. Experimental Design

Treatments (T) were set under two management conditions on two cultivars: two stems (T1) and three stems (T2) on a 'Cannon' cultivar and two stems (T3) and three stems (T4) on a 'Bragi' cultivar. It was a two-factor test. The area of each treatment was 26.5 $m^2$, with a total area of 106 $m^2$. The experiment was arranged on a complete factorial block design with three replicates and dimensions of 6 $m^2$. All data were subject to an appropriate analysis of variance.

Treatments mean differences were separated using Tukey's least significant difference (LSD) test at $p \leq 0.05$, and standard deviation was also calculated. Both procedures were estimated using MINITAB® Statistical Software (State College, PA, USA).

### 2.4. Determination of Biochemical Components

Total soluble solids (TSS °Brix) were determined by taking fruit juice, which was then placed on a digital refractometer Pr-100 (Atago®, Guangzhou, China) [10]. The pH was measured with 10 g of pulp and diluted in 50 mL of deionized water. Then, the solution was filtered to remove tissue remains. The pH was determined in an aliquot of 5 mL, using a potentiometer 12 model (CORNING, Corning City, NY, USA) [10]. Titratable acidity (TA) was measured by homogenizing 10 g of pulp in 50 mL of deionized water, then an aliquot of 10 mL was neutralized with (NaOH) at 0.1 N, adding phenolphthalein as an indicator [10]. The results were reported as a percentage of citric acid.

Vitamin C (ascorbic acid) was analyzed by homogenizing 20 g of fresh tissue in 30 mL of oxalic acid solution (0.5%), then an aliquot of 5 mL was titrated with 2,6-Dichloroindophenol solution (tilma solution) (0.01%). The concentration was expressed in $(100 \text{ g}^{-1} \text{ mg})$, using ascorbic acid as the standard [10]. Maturity index (MI) was obtained with the ratio of total soluble solids (TSS) (°Brix) and titratable acidity (TA). Lycopene was determined by colorimetry, with a reflection colorimeter D25A model (HUNTER Lab Virginia, Sunset Hills Rd, Reston, VA, USA) which was previously calibrated in order to obtain reliable results [11].

### 2.5. Yield and Fruit Size

The yield was determined with the weight of the fruits of eight plants for each treatment, as well as the number and size of fruits.

### 2.6. Determination of Postharvest Parameters

Firmness is one of the most important postharvest parameters to define shelf-life of fruits. This variable was individually measured on opposite sides of the diameter of five fruits per treatment and cultivar. Afterwards, a digital texture analyzer with a conical strut was used to perform the readings (FVD-30 model, Wagner Instruments, Greenwich, CT, USA). The color of the fruit was measured on the epidermis of the equatorial zone using a reflection colorimeter D25A model (HUNTER Lab Virginia, Sunset Hills Rd, Reston, VA, USA). The roundness index was determined with a digital vernier 9792 model (Hong Kong, China) and was expressed as the ratio of polar diameter (pd) and equatorial diameter (ed) of each fruit.

## 3. Results and Discussion

### 3.1. Biochemical Components

According to the data, TA, TSS, VC and MI did not show significant differences between treatments and cultivars. Citric acid content in fruits of 'Cannon' had the highest content (0.67%) on both treatments as compared to treatments of 'Bragi' (Table 1). This value was similar to that found by [2] on 'Triple 4', grown under greenhouse conditions. This value is also higher than the value of 0.25% found by [11] in the bell pepper 'Herminio'. This parameter is important because it is related to the taste of fruits; fruits with high values of citric acid have better taste.

**Table 1.** Content of titratable acidity (TA), total soluble solids (TSS), vitamin C (VC), maturity index (MI), pH, electrical conductivity (EC) and lycopene for bell pepper 'Cannon' and 'Bragi'.

| Treatments | TA (%) | TSS (°Brix) | VC (mg $100 \text{ g}^{-1}$) | MI | pH | EC (dS m$^{-1}$) | Lycopene ($100 \text{ mg}^{-1}$) |
|---|---|---|---|---|---|---|---|
| Cannon Two stems (T1) | 0.67 a | 8.85 a | 119.92 a | 14.27 a | 5.04 a | 1.38 a | 22.38 ab |
| Cannon Three stems (T2) | 0.67 a | 8.03 a | 120.23 a | 12.39 a | 5.03 a | 1.54 a | 23.54 a |
| Bragi Two stems (T3) | 0.47 a | 7.63 a | 120.42 a | 16.26 a | 5.00 a | 1.45 a | 11.84 c |
| Bragi Three stems (T4) | 0.51 a | 8.15 a | 120.15 a | 16.06 a | 4.99 a | 1.53 a | 14.06 bc |
| CV (%) | 18.33 | 6.25 | 0.17 | 12.27 | 0.47 | 5.04 | 1.38 |

Note: Different letters in each column indicate significant differences ($p \leq 0.05$).

Although there were no significant differences for TSS, the fruits of 'Cannon' showed a higher numerical value for this variable (8.85 °Brix) on treatments with two stems per plant (T1) when compared to treatments with 'Bragi' (Table 1). This variable indicates the amount of carbohydrates contained in the fruit as well as the concentration of dissolved minerals. Another article [2] found similar values (9.5 °Brix) on 'Viper' and 8.1 °Brix for 'California'. These values are higher than those reported in other species such as tomato [12].

Regarding VC content, all treatments had concentrations of 120 mg 100 g$^{-1}$ (Table 1). The levels found in this study are lower than concentrations of 274.3 and 355.5 mg 100 g$^{-1}$ reported by [2] on 'Viper' and 'Triple Star', respectively. The values found for this parameter is a clear indication of the high amount of VC contained in bell pepper as compared to the values of other fruits mentioned by [7–13].

The MI values were numerically higher on 'Bragi' fruit than those given treatments on 'Cannon'. An average value of 6.16 was obtained on 'Bragi', and 13.33 on 'Cannon', for both treatments. This index represents the quality of fruits in terms of taste and shelf-life. In addition, this variable is related to firmness (Table 1). Ref. [12] reported similar values for greenhouse tomato. The variables that showed the lowest values in terms of coefficient of variation (CV) were VC, pH and lycopene content, and those with the most variation in data were the AT and MI values.

Values of pH, EC and lycopene content were not statistically different between cultivars and treatments, and the interaction effects of these factors are shown in Table 1. The decrease in pH values is attributed to the variation in organic acid content in ionized form in plant tissue. The average value of EC (1.46 dS m$^{-1}$) was found on 'Cannon' and 1.49 dS m$^{-1}$ on 'Bragi' for both treatments. These values were close to those found by [14] for the same crop.

A higher concentration of lycopene was found in the fruits of 'Cannon' on treatments of two and three stems per plant. This process was attributed to the red fruits (Table 1). Nonetheless, these values agree with those reported by [15] on tomato. [2] reported lower values ranging from 0.092 to 0.66 mg 100 g$^{-1}$ in fruits of different varieties of greenhouse-grown bell pepper.

The concentration of lycopene is related to the ripening of fruit due to the increase in this carotenoid and the consequent decrease in chlorophyll when the fruit changes from green to red. In addition, bell pepper is one of the most important vegetable crops due to its high vitamin C content, carotenoids, phenols, capsaicinoids, xanthophylls and flavonoids [16]. It has been found in some other research that tomato and its products are the main source of lycopene in the human diet. Approximately 85% of lycopene that humans consume comes from this vegetable, while the rest comes from apricots, pink grapefruit, watermelon, guava and papaya [17].

Another important source of lycopene comes from red pepper [18]. Red pepper has nine times more lycopene than green peppers [19]. The concentration of lycopene content in bell pepper fruits depends on several factors that include variety, method of sampling, preparation, determination method, natural variation between fruits, fertilization, climatic conditions, soil properties, solar radiation, geographical origin, postharvest conditions and others [20]. The interest in carotenoids, particularly lycopene, has grown exponentially due to studies that suggest its impact on health and human diseases. Lycopene is not toxic and has antioxidant, anti-inflammatory and chemotherapeutic effects in heart and/or brain diseases and on some types of cancer [21].

Due to lycopene not being converted into vitamin A, it can be readily available for other properties (antioxidation), and the lack of ring structure of bioionone for lycopene can increase its antioxidant activity. These properties are much different to others, such as carotenoids consumed, making it present in its unique form in specific cellular tissues [22].

*3.2. Variation in Firmness and TSS at Different Stages of Maturity*

Figure 1 shows the variation in firmness at different stages of ripening in pepper fruits. The maximum value was presented on dark green stage for both treatments. The average

value (5.25 N) was compared with the fruits obtained in the ripening stage (3.69 N); this stage represented the optimal time for harvesting. Overall, it was observed that as the ripening state progressed, the firmness of the fruits decreased, in contrast to the increase in TSS content. Therefore, firmness is a variable that represents the consistency and texture of the fruits, which are attributes found in vegetables to establish the optimal time for harvest, quality evaluation during storage, marketing in the fresh state or processing. Furthermore, Ref. [7] reported that firmness of bell pepper is directly related with maturity at harvest. The consistency of fruits is reduced if fruits are harvested with their specific color of the variety, which affects the management and transport, thus decreasing shelf-life [9].

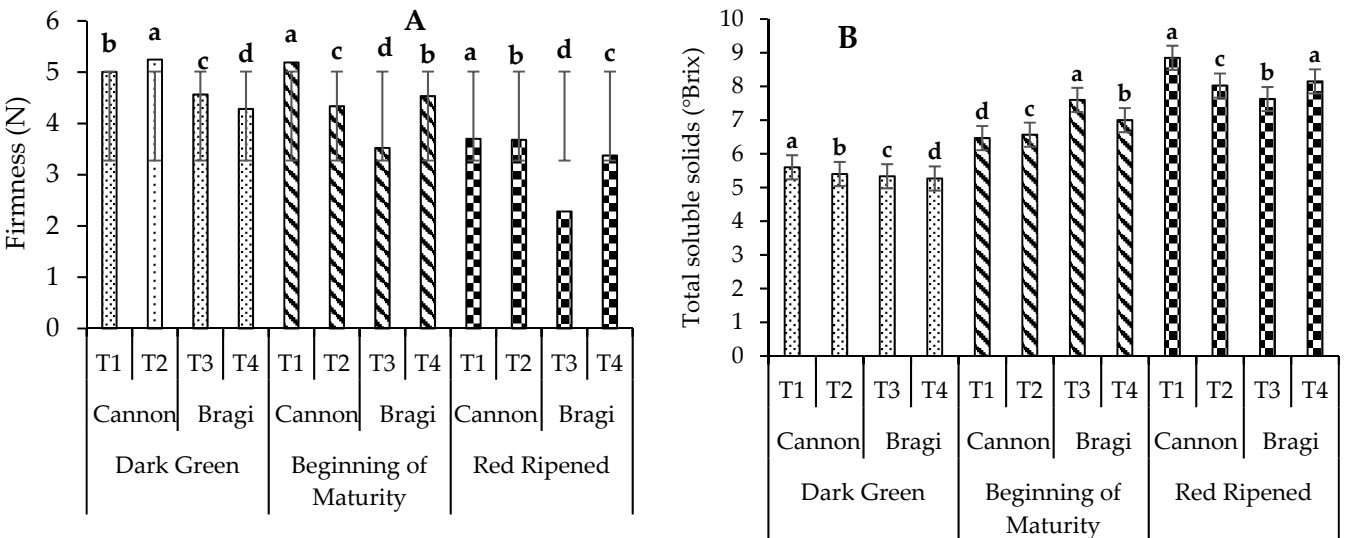

**Figure 1.** Variation in firmness (**A**) and total soluble solids (**B**) at different ripening stages in greenhouse-grown bell pepper. Different letters on each bars indicate significant differences ($p \leq 0.05$).

TSS: The maximum value of TSS was found in the fruits harvested in the ripening stage (red), which is commonly known as (consumption maturity), with a value of 8.85 °Brix for T1 (two stems) on 'Cannon'. When the fruits are harvested before their ripening point (dark green), they have less accumulation of sugars (°Brix). The optimal time for harvest or cutting of fruits is when they show their change in color (also known as harvest maturity). Accurate values of firmness and °Brix are presented for marketing purposes, as shown in Figure 1.

### 3.3. Physical Properties and Fruit Color

Table 2 shows the physical characteristics of bell pepper, such as weight of fresh fruit, roundness index and firmness. In the variables of fresh weight (FW) and roundness index, no significant differences were found between cultivars and/or treatments. The exception to this was firmness, where the highest average values (4.09 N) were found on 'Cannon'. A much lower value (12.64 and 21.60 N) is reported by [2] for bell pepper. Ref. [15] also reported values of 6.39 and 7.0 N firmness for the same crops—'Fascinato' (red bell pepper) and 'Jeanette' (yellow bell pepper)—grown in shade greenhouse conditions.

Shelf-life of fruits is characterized for its value as a quality and postharvest parameter [13]. Therefore, high values of firmness are desirable for those products that must be transported over long distances before reaching a consumer's market. Furthermore, it was found that fruits on T2 and T4, with three stems per plant, had smaller fruits on both cultivars as compared to fruits on T1 and T3, that had higher greater weight.

**Table 2.** Physical properties and color of pepper fruits for 'Cannon' and 'Bragi' cultivars in greenhouse black pepper treatments.

| Treatments | Physical Properties | | | Lycopene Values | | |
|---|---|---|---|---|---|---|
| | FW (g) | RI | Firmness (N) | Luminosity (L) | Purity (Chroma) | Hue |
| Cannon Two stems (T1) | 214 a | 0.88 a | 4.09 a | 24.33 a | 22.46 a | 63.75 a |
| Cannon Three stems (T2) | 198 a | 0.72 a | 3.92 a | 26.14 a | 26.41 ab | 63.46 a |
| Bragi Two stems (T3) | 240 a | 0.80 a | 2.43 ab | 38.18 b | 30.19 ab | 58.87 a |
| Bragi Three stems (T4) | 198 a | 0.81 a | 3.58 b | 39.26 b | 34.40 b | 58.41 a |
| CV (%) | 9.34 | 8.31 | 21.32 | 24.33 | 22.46 | 63.75 |

Note: Different letters in each column indicate significant differences ($p \leq 0.05$).

Values of fresh weight of fruits on this work are higher than those found by [4], which reported on 150–160 g of the varieties 'Orión' and 'Triple Star' harvested in the 3rd and 4th bifurcation of the plant. According to the results obtained, the weight and size of fruits vary as maturity progresses; because growth begins once the fruit is formed, if harvested on immature stage (green), the overall weight and size will decrease.

It was also observed square-shaped fruits, on both cultivars with a ratio between the length and width < 1. The properties of fruit color showed higher CV compared to physical variables such as fresh weight (FW), roundness index (RI), and firmness (Table 2).

*3.4. Lycopene Value*

The color evolution of fruit during its development was not evaluated but was measured only at harvest, since this crop is considered non-climacteric. Table 2 shows that red fruits on T1 and T2 of 'Cannon' exhibited less brightness than yellow fruits, as well as lower purity (chroma). Fruits of 'Bragi' presented hue values of 58.87 and 58.41 for T3 and T4, which are associated with lower lycopene content. While 'Cannon' fruits on T1 and T2 presented higher hue values (63.75 and 63.46). This process is related to a higher lycopene content [2–15].

On the other hand, Ref. [23] found values of 35.9 in the angle of luminosity (L), 44.8 for purity (a) and 24.46 for hue (b). Thus, the color of fruits plays an important role in consumer demand.

The quality of fresh fruits includes freshness [24]. Freshness includes a bright appearance in a variety of fruits [25]. Brightness makes the fruit attractive to the naked eye. Very often, the retail market and the fresh fruit market request a precise determination of freshness [26,27].

*3.5. Yield and Fruit Size Classification*

In this study it was found that T4 on 'Bragi' (yellow pepper) reached the highest yield (6.50 kg m$^{-2}$), with a total of 15 fruits per plant (Figure 2). Overall, treatments with three stems per plant in both cultivars reached better yields as compared to treatments with two-stems. However, no significant differences were found between cultivars and/or treatments. In that respect, some research [28] reported yields of 7.44 and 7.62 kg m$^{-2}$ on 'Magno' (oranged color) and 'Dicarpio' (yellow color), grown with tezontle under hydroponic system. Ref. [4] also found that yield and quality of fruits decrease as plant density also increase. They reported yield values of 5.25, 4.05 and 4.48 kg m$^{-2}$, with densities of 5, 6.5 and 8 plants m$^2$. These results agree with those reported by [6], who found that yield and size of tomato fruits decreased as the number of stems per plant increased. These results show that number of fruits increased with the number of stems per plant, but that the size of fruit decreased correspondingly.

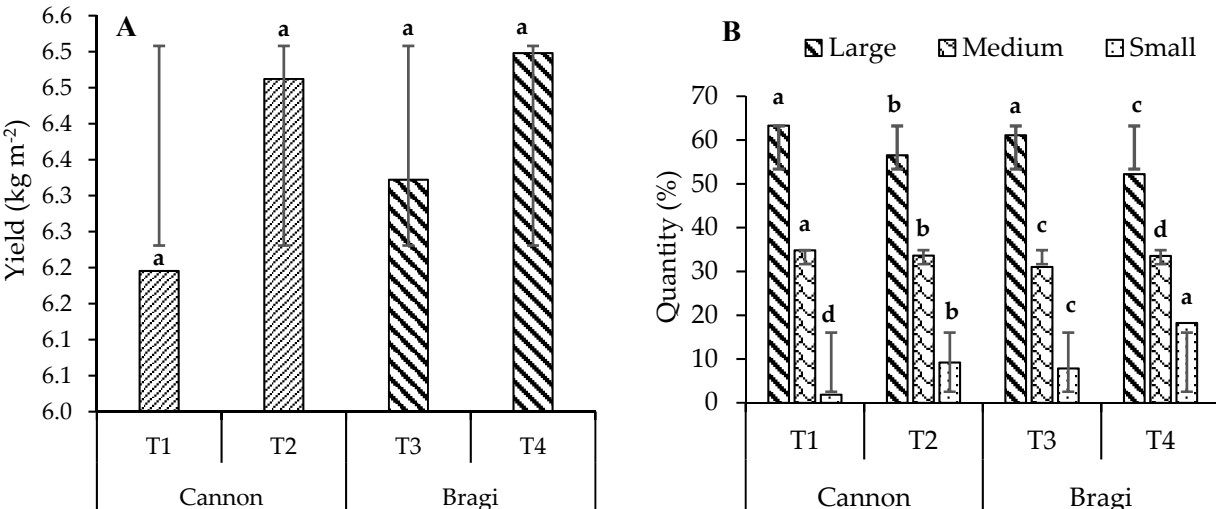

**Figure 2.** Yield (**A**) and size (**B**) of black pepper fruits. Different letters in each column indicate significant differences ($p \leq 0.05$).

In addition, T1 on 'Cannon' reached 63% of fruits for foreign market (large size), 35% for domestic market (medium size fruits) and 2% fruits for local market (small fruits) (Figure 2). Ref. [29] obtained 89% fruits for export, 4.39% for domestic market and 5.65 laggard fruits. Ref. [30] obtained 75% of first-class fruits for export on CV Anaconda grown on a macro tunnel system.

Ref. [31] reported higher percentages of extra-large fruits when applying fruit thinning as a cultural practice. Approximately 12 and 14 fruits were obtained on 'Cannon', and 12 fruits for T3 and 15 for T4 on 'Bragi', respectively.

### *3.6. Future Perspectives*

One of the future investigations will be to establish an experiment with application of different types of biostimulants in different pepper varieties with the purpose of improving quality and yield of fruit.

### 4. Conclusions

Modern horticultural techniques were used as a system of crop production in order to test the performance of bell pepper to obtain better organoleptic characteristics. The modification of the stem in the pepper crop induces the remarkable physiological characteristics in the fruit, both in its phenotype and in its taste properties.

It is recommended that growers use the two-stem-per-plant system to obtain larger fruits for export and to boost crop health. Increasing the number of stems per plant increases the total number of fruits, but the fruit size decreases. The amount of lycopene in bell peppers is an important variable that has gained interest for its antioxidant properties, but its concentration depends on the maturity stage and management of the production system.

**Author Contributions:** Conceptualization, C.M.-P. and J.F.-V.; methodology, C.M.-P. and J.F.-V.; software, J.E.R.-P.; validation, C.M.-P., J.F.-V. and J.d.R.R.-I.; formal analysis, C.M.-P.; investigation, C.M.-P.; resources, J.E.R.-P.; data curation, J.d.R.R.-I.; writing—original draft preparation, C.M.-P.; writing—review and editing, C.M.-P.; visualization, J.F.-V.; supervision, C.M.-P. and J.F.-V. All authors have read and agreed to the published version of the manuscript.

**Funding:** This research received no external funding.

**Data Availability Statement:** Not applicable.

**Acknowledgments:** A special thanks to Graduate College for providing the facilities to establish the experiment and the lab to assure the analysis of the variables studied.

**Conflicts of Interest:** The authors declare no conflict of interest.

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
