# Peer review of "Quality and Yield of Bell Pepper Cultivated with Two and Three Stems in a Modern Agriculture System"

_horticulturae, doi:10.3390/horticulturae8121187_

Round 1

Reviewer 1 Report

The manuscript title “Quality and yield evaluation of bell pepper cultivated with two and three stems in a modern agriculture system” highlighted on the quality and yield of bell pepper under hydroponic system. This MS need minor improvements.

Reviewer comments:

1-      In abstract, first introduce the crop and then raise the scientific question that need to be answered, don’t directly start results.

2-      Standard error or Standard deviation haven’t presented in all results figures and tables… Why? I suggest author to add them.

3-      Line 49: “Lycopene is a carotenoid mainly found in tomato (Solanum lycopersicum L.), bell pep- 49 per (Capsicum annuum L.) and watermelon (Citrullus lanatus var. Lanatus) and other fruits 50 or vegetables [6-7]. It also has antioxidant, anti-inflammatory and chemotherapeutic ef- 51 fects over heart-brain diseases and cancer”….. Lycopene has significant antioxidant property, why authors didn’t measure the antioxidant activity/capacity in this study. If possible than add the antioxidant activity results; it will enhance the significance of MS.

4-      In conclusion line 254-256 is not your research conclusion! It is introduction/ discussion part. I suggest authors to remove this and add your result conclusions….

Author Response

Dear Reviewer 1

All of the observations were corrected Please see the attachment

Reviewer 2 Report

All comments and corrections are inserted in text.

Author Response

Dear Reviewer 2

Many thank you for your kind observationes, all of them were corrected. Please see the attachment

Round 2

Reviewer 2 Report

Thank you fpr the acceptance of corrections.

Author Response

Dear Reviewer
